# Determinants of intention to leave among nurses and physicians in a hospital setting during the COVID-19 pandemic: A systematic review and meta-analysis

Neeltje de Vries[1,2], Laura Maniscalco[3], Domenica Matranga[3], José Bouman[2], J Peter de Winter[2,4,5,6]*

1 Department of Internal Medicine, Spaarne Gasthuis, Haarlem and Hoofddorp, the Netherlands, 2 Spaarne Gasthuis Academy, Spaarne Gasthuis, Haarlem and Hoofddorp, the Netherlands, 3 Department of Health Promotion, Mother and Child Care, Internal Medicine and Medical Specialties, "G. D'Alessandro" (PROMISE), University of Palermo, Palermo, Italy, 4 Department of Paediatrics, Spaarne Gasthuis, Haarlem and Hoofddorp, the Netherlands, 5 Leuven Child and Health Institute, KU Leuven, Leuven, Belgium, 6 Department of Development and Regeneration, KU Leuven, Leuven, Belgium

* pdewinter@spaarnegasthuis.nl

## Abstract

### Background

The global outbreak of COVID-19 has brought to light the profound impact that large-scale disease outbreaks can have on healthcare systems and the dedicated professionals who serve within them. It becomes increasingly important to explore strategies for retaining nurses and physicians within hospital settings during such challenging times. This paper aims to investigate the determinants of retention among nurses and physicians during the COVID-19 pandemic.

### Method

A systematic review of other potential determinants impacting retention rates during the pandemic was carried out. Secondly, a meta-analysis on the prevalence of intention to leave for nurses and physicians during the COVID-19 pandemic.

### Findings

A comprehensive search was performed within four electronic databases on March 17 2023. Fifty-five papers were included in the systematic review, whereas thirty-three papers fulfilled the eligibility criteria for the meta-analysis. The systematic review resulted in six themes of determinants impacting intention to leave: personal characteristics, job demands, employment services, working conditions, work relationships, and organisational culture. The main determinants impacting the intention to leave are the fear of COVID-19, age, experience, burnout symptoms and support. Meta-analysis showed a prevalence of intent to leave the current job of 38% for nurses (95% CI: 26%-51%) and 29% for physicians (95%

**Data Availability Statement:** All relevant data are within the manuscript and its Supporting Information files.

**Funding:** The author(s) received no specific funding for this work.

**Competing interests:** The authors have declared that no competing interests exist.

**Abbreviations:** COVID-19, Coronavirus disease; WHO, World Health Organization; PRISMA, Preferred Reporting Items for Systematic Review and Meta-Analyses; PROSPERO, International Prospective Register of Systematic Reviews; MMAT, Mixed Methods Appraisal Tool; PTSD, Post-Traumatic Stress Disorder; PMIEs, Perpetrate Potentially Morally Injurious Events; PPE, Personal protective equipment.

CI: 21%-39%), whereas intention to leave the profession for nurses 28% (95% CI: 21%-34%) and 24% for physicians (95% CI: 23%-25%).

## Conclusion

The findings of this paper showed the critical need for hospital managers to address the concerning increase in nurses' and physicians' intentions to leave during the COVID-19 pandemic. This intention to leave is affected by a complex conjunction of multiple determinants, including the fear of COVID-19 and the confidence in and availability of personal protective equipment. Moreover, individual factors like age, experience, burnout symptoms, and support are maintained in this review. Understanding the influence of determinants on retention during the COVID-19 pandemic offers an opportunity to formulate prospective strategies for retaining nurses and physicians within hospital settings.

## Introduction

On March 11, 2020, the outbreak of the coronavirus disease (COVID-19) was declared by the World Health Organization (WHO) as a global pandemic [1]. The COVID-19 outbreak increased the short-term inpatient load, resulting in a lack of hospital capacity [2, 3]. Hospitals tried to manage the increase in hospitalisations by cancelling the admission of regular care [3]. To overcome the extra needed care of COVID-19 patients, nurses and physicians were stretched during the pandemic [4]. Once more, nurses and physicians showed they played a crucial role in maintaining patient care in hospitals [5, 6].

However, global healthcare has already struggled with the shortage of healthcare workers in the past decade. For the future, the WHO predicts a shortage of 18 million healthcare workers worldwide in 2030 [7]. More specifically, 6.4 million physicians are needed to face the shortage [8]. This certainty is the result of cumulative issues. Firstly, the shortage is driven by a large number of chronically ill people and the increasing demand for healthcare [9]. Second, life expectancy is increasing consistently and will increase morbidity and age-related diseases [10]. Lastly, the outflow of nurses and physicians will increase due to the retirement of ageing healthcare personnel. Half of the physicians who practised in 2021 and 20% of the nurses [11] will be retired by 2030 [12, 13]. In summary, the healthcare workforce is vulnerable and employed nurses and physicians should be maintained as much as possible.

After that, the COVID-19 pandemic put severe pressure on the already vulnerable healthcare workforce. Maleki et al. (2021) described that pandemics, like COVID-19, can put a severe burden on the healthcare system [14]. Nurses and physicians were exposed to acute stress [14–16] and high workloads [16]. Previous research showed that workload, stress and burnout symptoms increase the intention to leave [17]. It is essential to focus on retaining healthcare personnel. In contrast, nurses and physicians leaving their jobs can create severe financial and non-financial burdens to the healthcare system [18], e.g. unsatisfactory quality of care [19], recruiting costs [20–23] and dissatisfaction with patients and personnel [20, 24, 25]. Especially during this pandemic crisis, the burden on nurses and physicians was severe [14]. Future pandemics or epidemics can significantly impact the healthcare system and, thus, the outflow of healthcare workers. Moreover, it is crucial to understand how hospital managers can manage to retain their nurses and physicians during future outbreaks.

The likelihood of another pandemic impacting the healthcare workforce is strong. In the last century, the occurrence of disease outbreaks has risen [26] as a result of a growing population, intensified global interconnectedness, microbial adjustments and variations, the development of the economic situation, land utilization and climate change [27, 28].

Knowing the impact of the determinants on retention during COVID-19, it is possible to create an upcoming policy for retaining nurses and physicians in healthcare during possible future pandemics or other moments of crisis. Similar studies have been published in the past years. Nevertheless, these papers, among other things, focused on nurses only [6, 29–32], measured mental issues [29, 33, 34] increasing the intention to leave, examined the impact of viral outbreaks instead of COVID-19 [35] or lacked a systematic approach [36]. In summary, this paper has indicated added value to studying the impact of COVID-19 on the intention to leave nurses and physicians in a hospital now that the COVID-19 pandemic has ended [37]. The results of this paper can be used creating policies to retain nurses and physicians in a hospital setting, particularly in times of crisis. Two-fold goals set the aim of this study. They are first conducting a systematic review of potential determinants impacting the retention rates during the pandemic and secondly, performing a meta-analysis of the prevalence estimates of the intention to leave for both nurses and physicians during the COVID-19 pandemic.

## Materials and method

The systematic review and meta-analysis were carried out in accordance with the Preferred Reporting Items for Systematic Review and Meta-analysis (PRISMA) [38]. The PRISMA Checklist is available in S1 Table. The systematic review was registered at the International Prospective Register of Systematic Reviews (PROSPERO), CRD42023403693.

### Literature search

The authors developed an extensive search string for three scientific databases: PubMed, Embase, CINAHL, and Web of Science. The search was framed using the domain (nurses and physicians), exposure (COVID-19) and outcome (job retention/turnover) within the last ten years. Synonyms used for building the search were: **'Health Personnel' [Mesh], 'Health Personnel', 'healthcare professional*', 'health care professional*', 'healthcare worker*', 'health care worker*', 'healthcare provider*', 'health care provider*', 'Healthcare staff', 'Health care staff', and 'Health worker*', 'Nurses' [Mesh], 'nurse', 'nursing', 'nurse's role'[Mesh], 'Nursing Staff, Hospital' [Mesh], 'Physicians'[Mesh], 'Physician*', and 'medical specialist'.** 'COVID-19' [Mesh]), 'Covid-19','Covid-19', 'SARS-CoV-2' [Mesh], 'SARS-CoV-2', 'COVID-19 pandemic', and 'severe acute respiratory syndrome coronavirus 2' were synonyms used to study exposure. Moreover, the search was amplified by the following synonyms for the outcome frame: 'Personnel Turnover' [Mesh], 'Personnel Turnover*', 'Employee Turnover*', 'Personnel Retention*', 'job retention', 'retention rates', 'turnover intention*', 'intention to leave', 'intention to quit', 'intent to quit', 'intent to leave', 'staff turnover*', 'quit the job', 'retaining personnel', 'intention to stay', and, 'retention'. The complete search string of the three scientific databases can be found in S2 Table. The study search was conducted in the week of March 17, 2023.

### Inclusion and exclusion criteria

Quantitative research and mixed methods were included in the manuscript. Only the quantitative part of the mixed-method studies will be included. Further inclusion criteria were addressing nurses and physicians, being carried out in a hospital setting during the COVID-19 pandemic and being written in English. Papers were excluded if full text was not available.

Moreover, qualitative studies, systematic reviews, study protocols, and dissertations were excluded.

### Screening and data extraction

A pair of independent reviewers (NdV, LM) screened the title and abstract for inclusion and exclusion criteria. After this first screening, the remaining papers were screened on inclusion and exclusion criteria by reading the full text. Disagreement about study eligibility was resolved through consensus discussion or by an extra author, not a member of this duo.

A data extraction sheet was developed to extract relevant information, including the type of study, country, sample and sample size, prevalence outcomes, measurement, measurement method, and determinants. In addition, a quality assessment was executed by two independent reviewers (Ndv, LM) to reduce the risk of bias. Because of the possibility of heterogeneity of the included papers, the Mixed Method Appraisal Tool (MMAT) was used. Disagreement about the outcomes was resolved through consensus discussion or by an extra author, not a member of this duo.

### Meta-analysis methodology

The meta-analysis was based on the screened papers that reported the prevalence estimates of the intention to leave the current job or the profession for both nurses and physicians. The data were analysed using the Metaprop package, a statistical tool within the Stata software designed to conduct meta-analyses of proportions [39]. The pooled estimate was obtained utilising fixed or random effects models. The random-effects model was based on the DerSimonian and Laird method [40]. The model was reduced to a fixed-effects model when the variance parameter that describes the heterogeneity was zero. The inverse variance-weighted method was used for the fixed effects model, where each study estimate was given a weight directly proportional to its precision. Forest plots represented the estimated effect of each study and were presented with 95% exact binomial Clopper-Pearson confidence intervals (95%CI) [41]. The Freeman-Tukey double arcsine transformation was used to stabilise the variance of the pooling proportions. This transformation avoided excluding studies with a proportion of 0 or 1, where the standard error would be zero and weights would be infinite. Statistical heterogeneity was assessed by $I^2$ [42], with a higher value indicating higher inter-study heterogeneity. P-values <0.05 were considered to be statistically significant.

## Results

### Study selection process

The literature search resulted in 2261 papers. Before the screening, 674 duplicates were removed. After title and abstract screening, 1045 records were removed due to not fulfilling the inclusion criteria. The full text of 158 records was screened. Finally, 55 papers fulfilled the eligibility criteria and were included in the systematic review and of these, 33 papers could be contained in the meta-analysis. The study selection process is shown in the flow chart (Fig 1).

### Characteristics of the studies

The 55 papers in the systematic review included three descriptive studies, one experimental study, and one quasi-experimental study design. The others were cross-sectional study designs. Three of the included papers addressed physicians, 34 addressed nurses, and ten papers studied both nurses and physicians. There was heterogeneity in the measurement of the intention to leave. Eight studies measured the intention to leave the current job; four studies measured

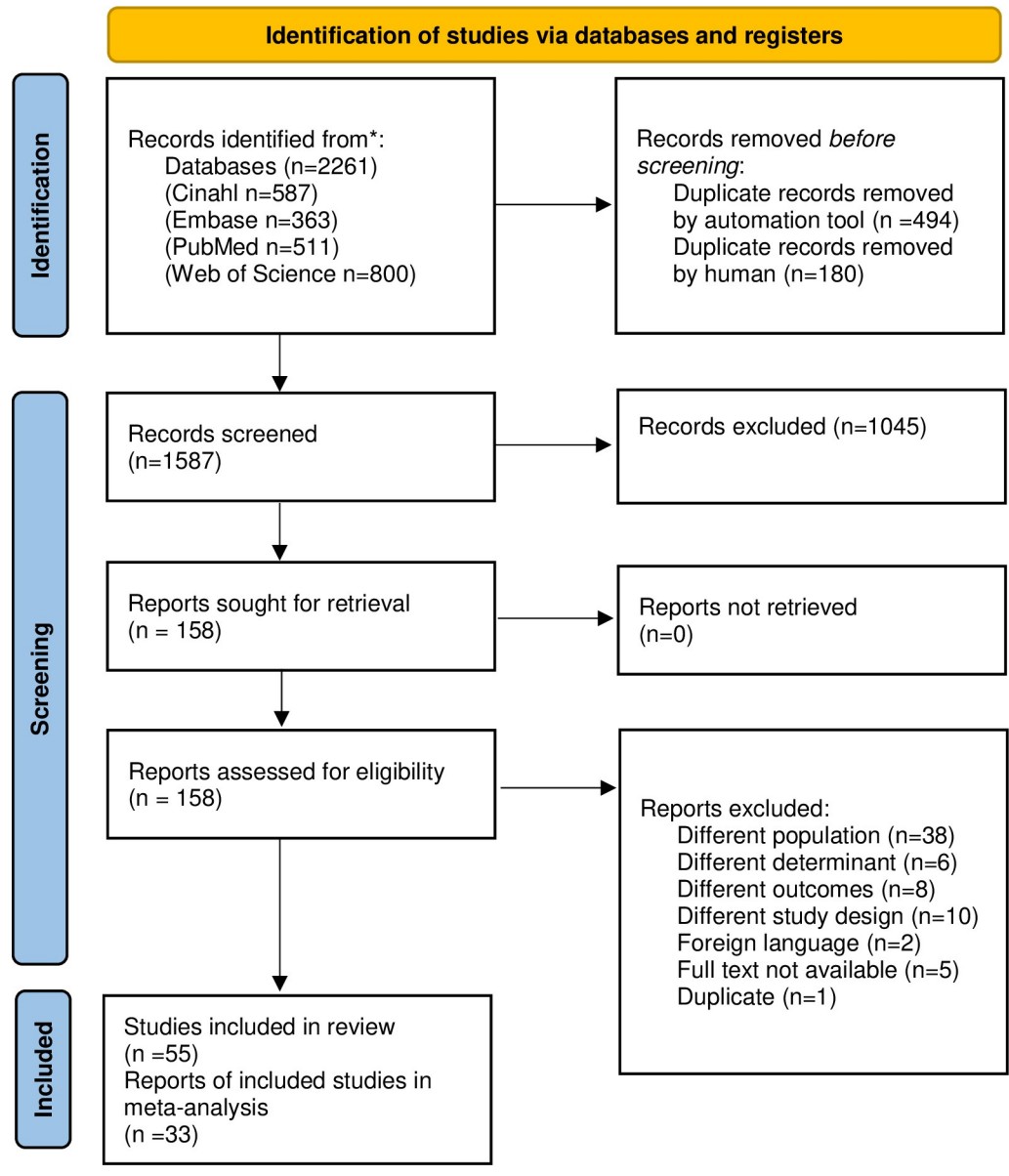

**Fig 1. PRISMA flow diagram.**

the intention to leave the profession, six studies measured both, one study determined the intention to quit, two studies measured the intention to stay, and 26 studies studied the intention to leave (not further specified). The extraction table, including quality assessment, can be found in S3 Table.

The study characteristics and quality assessment of the 3,329 studies selected for meta-analysis are shown in Table 1. Thirty studies use a cross-sectional study design. Three mixed-methods designs are included. Only the quantitative part with a cross-sectional design is used for analysis. Three papers addressed only physicians, 27 of the included documents addressed nurses alone, while three studies targeted nurses and physicians together. In terms of measurement of the intention to leave, 18 papers studied nurses' intention to leave their current job, 165 papers studied nurses' intention to leave the profession, two studies showed physicians'

**Table 1. Data-extraction table and quality assessment summary of included records in meta-analysis.**

| First Author (Year), Country | Type of study | Sample (n) | Prevalence | Measurement | Measurement method | Quality Assessment[a] |
|---|---|---|---|---|---|---|
| Alameddine (2021) [43], Lebanon | Cross-sectional | Nurses (362) | 156 | Intention to leave the current job | How likely are you to quit your current job in the next 1–3 years? | 5/5 |
| Alameddine (2021) [44], Lebanon | Cross-sectional | Nurses (265) | 74 | Intention to leave the current job | In the coming 12 months, do you intend to quit? | 4/4 |
| Alhenaidi (2023) [45], Kuwait | Cross-sectional | Physicians (89) | 48 | Intention to leave the current job | Intention to leave their current job as an ICU physician | 4/4 |
| Al-Mansour (2021) [46], Saudi Arabia | Cross-sectional | Nurses (242) Physicians (340) | 84 164 | Intention to leave the current job | Are you intend to leave your primary healthcare centers within the next few months? | 4/4 |
| Barshied (2023) [47], United States | Cross-sectional | Physicians (455) | 146 | Intention to leave the current job | Likely to seek a job within a year | 3/4 |
| Bruyneel (2023) [48], Belgium | Cross-sectional | Nurses (2183) | 519 936 | Intention to leave the current job Intention to leave the current profession | The intention-to-leave the profession in the next year was assessed dichotomously (yes or no). The intention-to-leave the hospital in the next year was assessed dichotomously (yes or no). | 3/4 |
| Christianson (2023) [49], United States | Cross-sectional | Nurses (1299) | 285 | Intention to leave the profession | Do you want to continue to work as a nurse? | 3/4 |
| Cornish (2021) [50], Australia | Cross-sectional | Nurses (398) | 192 82 | Intention to leave the current job Intention to leave the profession | The intention-to-leave the hospital in the next year was assessed dichotomously (yes or no). The intention-to-leave the profession in the next year was assessed dichotomously (yes or no). | 3/4 |
| Crowe (2022) [51], Canada | Mixed method | Nurses (425) | 92 | Intention to leave the current job | Intention to quit their current employment | 3/4 |
| Djupedal (2022) [52], Norway | Cross-sectional | Nurses (694) | 67 | Intention to leave the profession | Has the COVID-19 pandemic led you to consider quitting as a nurse when the crisis is over? | 4/5 |
| Falk (2022) [53], Sweden | Cross-sectional | Nurses (37) | 10 | Intention to leave the current job | Intention to leave their current position within 6 or 12 months | 3/4 |
| Hendrickson (2022) [54], United States | Cross-sectional | Nurses (161) Physicians (56) | 95 26 | Intention to leave the profession | How likely do you think it is that you will still be working in your current field in 5–10 years? | 2/4 |
| Kantorski (2022) [55], Brazil | Cross-sectional | Nurses (890) | 219 | Intention to leave the profession | Did you think about changing professions at some moment during the COVID-19 pandemic? | 3/5 |
| Kitamura (2022) [56], Japan | Cross-sectional | Nurses (84) | 49 | Intention to leave the current job | Do you have internal transfer intentions after the introduction of COVID-19 related family visiting restrictions? | 4/4 |
| Kleier (2022) [57], United States | Cross-sectional | Nurses (189) | 23 | Intention to leave the profession | Intention to remain in the nursing profession | 5/5 |
| Li (2020) [58], China | Cross-sectional | Nurses (1646) | 22 | Intention to leave the current job | Intention to leave due to COVID-19 | 4/5 |
| Li, Pien et al. (2022) [59], Taiwan | Cross-sectional | Nurses (1499) | 341 | Intention to leave the current job | Intention to leave their current job during the pandemic in comparison with that before the pandemic | 3/4 |
| Li, Qiu et al. (2022) [60], China | Cross-sectional | Nurses (1204) | 138 | Intention to leave the profession | Given the current situation, I am more likely to leave my profession | 4/5 |
| Nakai (2022) [61], Japan | Cross-sectional | Nurses (81) | 49 | Intention to leave the current job | I would like to quit my current job | 2/4 |
| Nakić (2023) [62], Croatia | Cross-sectional | Nurses (120) | 47 | Intention to leave the profession | Intention to leave the nursing profession | 4/4 |
| Nielsen (2022) [63], Denmark | Cross-sectional | Nurses (377) | 147 | Intention to leave the current job | The management of COVID-19 in my workplace has made me consider changing jobs | 5/5 |
| Özkan Şat (2021) [64], Turkey | Cross-sectional | Nurses (263) | 137 | Intention to leave the profession | Intention to leave profession during the COVID-19 pandemic | 3/4 |
| Petri☐or (2021) [65], Romania | Cross-sectional | Nurses (79) | 21 | Intention to leave the current job | Considering to resign the current position during the COVID-19 pandemic | 4/5 |
| Rhéaume (2022) [66], Canada | Cross-sectional | Nurses (178) | 87 | Intention to leave the current job | Have you considered leaving your employment as a result of COVID-19? | 4/4 |

*(Continued)*

**Table 1.** (Continued)

| First Author (Year), Country | Type of study | Sample (n) | Prevalence | Measurement | Measurement method | Quality Assessment[a] |
|---|---|---|---|---|---|---|
| Roberts (2022) [67], United Kingdom | Cross-sectional | Nurses (161) | 41 | Intention to leave the profession | Have you considered leaving nursing because of the pandemic? | 4/4 |
| Said (2021) [68], Egypt | Cross-sectional | Nurses (210) | 52 | Intention to leave the profession | Intent to leave current position, current organization, the field of nursing | 4/4 |
| Schug (2022) [69], Germany | Cross-sectional | Nurses (757) | 143 | Intention to leave the profession | Have you felt so stressed due to job-related factors in the last three months that you plan to quit your job within the medical field? | 4/4 |
| Sinsky (2021) [70], United States | Cross-sectional | Nurses (2302) Physicians (9266) | 921 2204 | Intention to leave the profession | What is the likelihood that you would leave your practice within 2 years? | 4/5 |
| Tamrakar (2023) [71], Nepal | Cross-sectional | Nurses (96) | 55 55 | Intention to leave the current job Intention to leave the profession | Did not want to continue working in the same department Quitting their job as nurses | 3/5 |
| Tang (2022) [72], China | Cross-sectional | Nurses (2480) | 1590 | Intention to leave the current job | Score obtained from Turnover Intention Scale | 5/5 |
| Wood (2021) [73], United Kingdom | Mixed method | Nurses (124) | 27 | Intention to leave the current job | Being more likely to leave their job now, than before the pandemic. | 4/4 |
| Wood (2022) [74], United Kingdom | Mixed method | Nurses (97) | 58 22 | Intention to leave the current job Intention to leave the profession | Leaving their current role Leaving nursing entirely | 3/4 |
| Yang (2022) [75], China | Cross-sectional | Physicians (3783) | 831 | Intention to leave the current job | Turnover intention of the current job in the next 12 months | 5/5 |

[a]The quality assessment was conducted using the Mixed Methods Appraisal Tool (MMAT) [76]. 'Yes' counted for one point and 'No' for zero points. In case a quality criterion was answered with 'cannot tell', more information was needed to give a legit answer in terms of 'yes' or 'no'[70]. Therefore, this criterion is not included in the overall score.

intention to leave their current job, and two studies examined physicians' intention to leave the profession.

### Review outcomes

**Personal characteristics.** Female participants had higher turnover intentions than their male colleagues [77–79]. Multiple studies described a significant association between turnover intention and age [79–83]. At the same time, younger nurses and physicians (<39 years old) [80, 81, 83] and aging workers showed higher intentions to leave [81, 84].

The impact of marriage is not homogenous in literature. Two studies described married nurses had higher scores of turnover intention [78, 85], whereas two other studies described that being married is associated with decreasing intention to leave [84, 86]. Being unmarried was suggested to impact the turnover intention[46,50] badly. Furthermore, having multiple children increases the turnover intention [82].

Results showed a significant correlation between emotional intelligence, job performance, and turnover intention. Higher emotional intelligence scores and above-average job performance scores lowered the intention to leave [79].

Lavoie-Tremblay et al. (2022) described that nurses who lacked professional competence and were overwhelmed at work during the pandemic showed higher intention to leave their current setting and the profession than their well-prepared colleagues [87]. Studies showed a positive relationship between professional health [78] and general health [85] and intention to

stay. Furthermore, an adverse effect was found between coping skills [88], self-disclosure [89] and turnover intention.

Finally, Rafiq et al. (2022) described life satisfaction or happiness as related to turnover intention [90].

**Job demands.** Burnout [89, 91–100] and burnout symptoms like depersonalisation [101, 102], emotional exhaustion [102–104], and personal accomplishment [102] showed a significant increase in intention to leave for nurses and physicians. Moreover, psychological issues like anxiety disorder or depression showed an association with the intention to resign [99]. Also, (chronic) fatigue was associated with the intention to leave the setting or the profession [87, 105]. Furthermore, turnover intention is predicted by job satisfaction [80, 87, 98, 106–108]. Also, job stress or stressful work is positively correlated with turnover intention [83, 103, 106, 109–111].

Heavy workload [77] and working night shifts [83, 112] or working rotating shifts [113] instead of only working dayshifts [82, 94] were predictive factors for considering leaving. Moral distress [100, 113–117], Post-Traumatic Stress Disorder (PTSD) [85, 118], and moral injuries are related to career abounded intention [104]. Khattak et al.(2021) described the correlation between turnover intention and secondary trauma [116].

Perpetrating potentially morally injurious events (PMIEs) and passively witnessing PMIEs during COVID-19 impacted turnover intention for nurses [97]. Two studies showed that increased scores of psychological resilience were associated with a lower score of professional turnover intention [93, 119]. In contrast, two other studies showed that resilience was negatively correlated with the intention to resign [106, 120].

Regarding COVID-19, higher COVID-19-associated discrimination is associated with higher scores on professional turnover intention [119], and higher turnover scores are shown for those who had a change of duty during the pandemic period [94]. Shah et al. (2021) demonstrated that state anger, in terms of how nurses experienced and felt about working as nurses during the COVID-19 outbreak [103], is correlated with the turnover intention of nurses [103]. When having pride in working as a critical professional, the odds of turnover intention were lower [93].

Jiwnani et al.(2021) showed that an increase in domestic responsibilities as a result of lockdowns during COVID-19 resulted in an increment in the intention to leave [121].

In terms of patient care, linear regressions explained that nurses taking care of male patients were less likely to have job turnover than their colleagues [98]. Moreover, turnover intention correlated with patient assignments [78] and compassion fatigue [122]. Furthermore, the lower the autonomy, competence, and relatedness satisfaction, the higher the turnover intention is [89]. At the same time, one study showed decision latitude as a predictor of turnover intention [85]. Mirzaei et al.(2021) summarised that the higher the job demands, job insecurity, or strain, the higher the turnover intention [85].

**Employment services.** The amount of working years [80] or working experience is related to turnover intention [81–84, 87, 123, 124]. Nurses who worked in the hospital for less than five years showing intention to leave scores than those working over six years in the hospital [77]. Gümüssoy et al.(2023) and Kim et al.(2020) showed that increasing lengths of service increased the intention to leave the profession [84, 94]. Chen et al.(2022) showed that healthcare workers who had worked for 5–10 years were a predictive factor for the intention to leave their job [83]. Moreover, Nashwan et al. (2021) described that participants in the range of 5–10 years of experience had a higher intention to leave during COVID-19 than all other groups of work experience [124]. Raso et al.(2021) described that nurses with >11 years of experience had lower turnover intention [81].

The amount of working hours per week is related to turnover intentions [82]. Working more than 50 hours a week increases the turnover intention [83].

Overall, Lou et al.(2021) showed that the availability of hospital resources is negatively correlated with the intention to quit [100]. They were being unsatisfied with their salary or having a low wage impacted negatively the turnover intention [82, 84, 94]. Also, the personal financial situation impacted the turnover intention; an increase in the economic situation decreased the intention to leave [98]. Moreover, the availability of job alternatives increases the turnover intention [125].

Finally, Hanum et al.(2023) described the importance of professional identity and the association with the intention to stay [80]. Also, development opportunities, like attending a programme to learn care during a pandemic [111], decrease the intention to leave.

**Working conditions.** Role [81], job title [80], professional title [82], and position [82, 84] are associated with intention to leave. Raso et al.(2021) showed that direct care nurses had the highest intention to leave [81]. Also, nurses working in the intensive care unit had higher turnover scores than colleagues working in other units [77, 114]. Zeng et al.(2022) confirmed that the work department is related to turnover intention [82]. Moreover, nurses who worked in the district had a higher intention to leave [77]. Additionally, the practice environment of nurses significantly predicted the intention to leave. Nurses who had a less friendly practice environment had a higher intention to leave [126].

Nashwan et al.(2021) showed that COVID-19 resulted in a higher turnover intention score compared to before the pandemic [124]. Several researchers studied the determinants of these higher turnover intention scores. Among other things, multiple studies showed turnover intention was correlated with the fear of COVID-19 [77, 107, 112, 116, 127] or coronaphobia [88]. Alnaeem et al.(2022) showed that the belief that nurses and physicians could die from COVID-19 more than from cancer was a predictive factor in considering leaving the job [112]. The same was found in taking care of confirmed or suspected COVID-19 patients [83, 87, 111] and for perceiving high risk of COVID-19 [83]. Likewise, greater intention to leave was reported in case nurses reported self-infection with COVID-19 or a team member got infected at work [87]. Literature shows that personal protective equipment (PPE) plays a significant role. For instance, a shortage of PPE [78, 91], a lack of confidence in the available PPE [91], and a lack of confidence in the PPE adequacy [91] were related to higher intention to leave.

Those who found the number of medical personnel insufficient at the unit had a higher intention to leave [94]. Likewise, a correlation was shown between intention to leave and patients' acuity, which is "the perceived degree of care and attention a patient's treatment requires" [78]. In regression analysis, the ability to take a rest [93] and the situation at work were significant determinants [87]. Finally, quality of work life correlates with turnover intention negatively [110].

**Work relationship.** Regarding work relationships, the literature showed that poor values alignment [109], poor teamwork [109], and labour relations [82] are related to turnover intention. Moreover, poor supportive administration or management [83] and getting along badly with the immediate leader negatively impact the intention to leave [82]. Furthermore, conflicts with colleagues or the supervisor were associated with an increase in turnover intention [93]. Organisational commitment is positively correlated with job retention [123], which is confirmed by Karimi et al.(2022); nurses who are less interested in the organisation had higher turnover intention than their colleagues [101]. Chen et al. (2021) showed the willingness to provide services with a positive impact on retention rates [111]. Moreover, high work engagement scores lowered the turnover intention [89].

During the COVID-19 pandemic, horizontal violence and bullying (person-oriented, work-related, and physically intimidating bullying) were correlated with turnover intention [128].

**Organisational culture.** Higher perceived organisational support showed decreased intention to leave in nurses and physicians [83, 85, 86, 93, 108, 129]. At the same time, less

appreciation of the organization is predicted, taking into consideration leaving the job [112]. Further, having supervisor support is a protective factor for turnover intention [89].

At the private level, having no support from family [82] or no psychosocial support badly impacts the turnover intention [78]. Moreover, two studies showed a negative correlation between social support and turnover intention [85, 88]. Leadership was found to have a significant impact on the turnover intention of nurses and physicians [87, 130]. Also, poor work control had an on turnover intention [109]. Rafiq et al.(2022) described a trusted climate lowering the turnover intention [90] as is the case of high macro-control perception, defined as "the participant's belief about the effectiveness of measures taken at the institutional and national level during the pandemic" in COVID-19 [129]. Organisational justice, measured as "How workers feel about their company's impartiality or fairness" [125], was positively correlated with turnover intention [125].

Lastly, job embeddedness showed a negative correlation with turnover intention [90].

**Prevalence of the intention to leave the current job.** For the nurses' sample, the minimum value was found in Li et al. (2020) paper with a prevalence of 1% (95% CI: 1%-2%) and the maximum was observed in Tang et al. (2022) with a prevalence of 64%: (95%CI: 62%-66%). The pooled prevalence of the intention to leave the current job for nurses calculated from 18 studies was equal to 38% (95%CI: 26%-51%) (Fig 2) with statistically significant intragroup heterogeneity (I$^2$ >99%, p<0.001).

For physicians, the authors found papers by Al-Mansour et al. (2021) showing a prevalence of 35% (95%CI: 29%-41%)by Barshied et al. (2023) with a prevalence of 35%, (95% CI: 28%-37%) and a paper by Yang et al.(2022) with a prevalence of 22%, (95%CI: 21%-23%). The pooled prevalence of the intention to leave the current job obtained from three studies was lower and equal to 29% (95%CI: 21%-39%) (Fig 3).

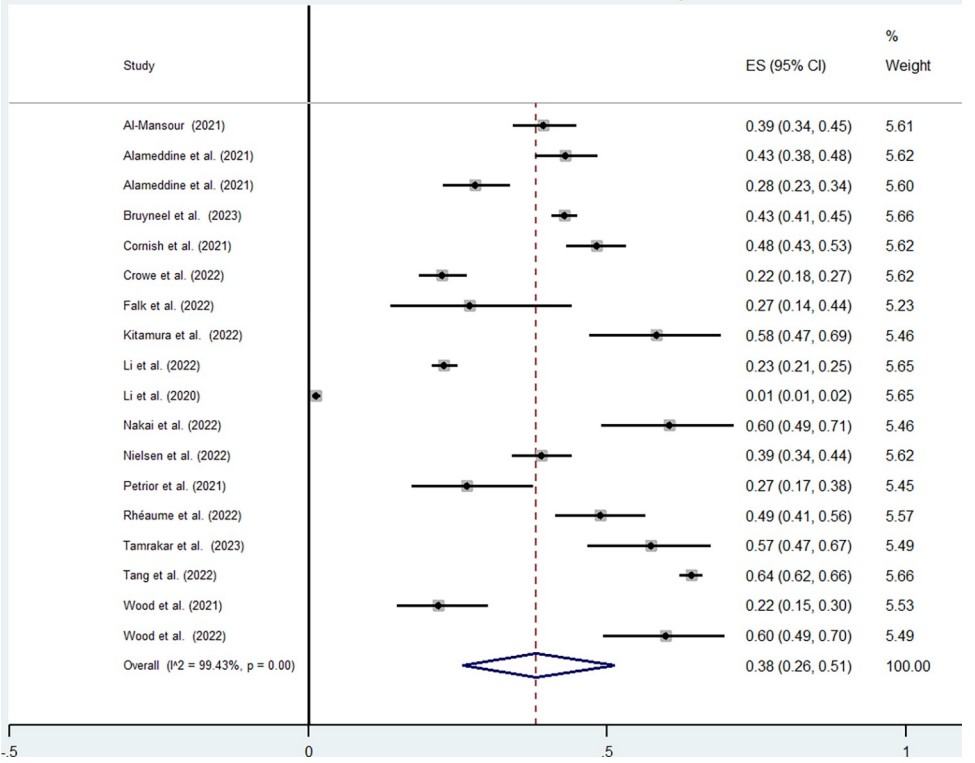

**Fig 2. Nurses' intention to leave the current job.**

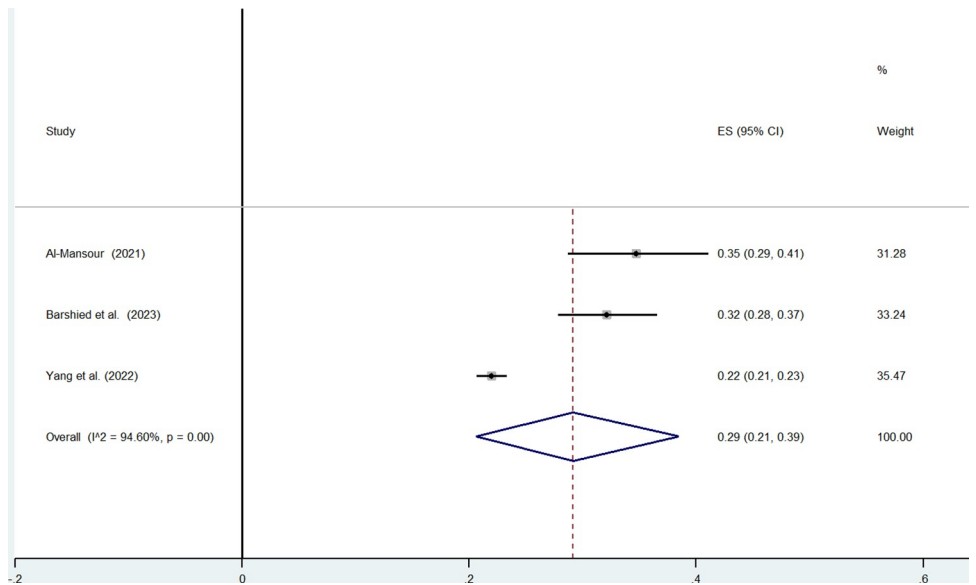

**Fig 3. Physicians' intention to leave the current job.**

**Prevalence of the intention to leave the profession.** For nurses, the minimum value was found in Djupedal et al. (2022) with a prevalence of 10%, (95%CI: 18%-12%) and the maximum was observed in Hendrickson et al. (2022) with a prevalence of 59%, (95%CI: 51%-67%). The pooled prevalence of the intention to leave the profession for nurses obtained from 16 studies was equal to 28% (95%CI: 21%-34%) with statistically significant intra-group heterogeneity ($I^2 > 98\%$, p<0.001) (Fig 4).

For physicians, Sinsky et al. (2021) showed a prevalence of 24%, (95%CI: 23%-25%) and a paper of Hendrickson et al. (2022) with a prevalence of 46%, (95%CI = 33%-60%). The pooled prevalence of the intention to leave the profession obtained from 2 studies was lower and equal to 24% (95%CI: 23%-25%) (Fig 5).

## Discussion

This study revealed a substantial increase in intention to leave among nurses and physicians during the COVID-19 pandemic. Nurses had a pooled prevalence of 37% for the intention to leave their current job and 27% for the intention to leave the profession. At the same time, physicians had rates of 33% for intention to leave their current job and 24% for intention to leave the profession respectively. The impact of determinants on intention to leave during the COVID-19 pandemic varied into six themes: personal characteristics, job demands, employment service, working conditions, work relationships, and organisational culture. The main determinants to be highlighted are fear of COVID-19 and lack of PPE, burnout symptoms, age, experience, and support.

An earlier meta-analysis on the intention to leave among nurses during the COVID-19 outbreak showed a prevalence of 31.7% [31] which is lower than our estimated intention to go during the pandemic. However, Ulupinar et al. (2022) calculated a prevalence of intention to leave the profession, In contrast, the estimated 37% in our paper is calculated using the intention to leave the profession and intention to leave the job. Accordingly, our paper showed lower prevalence in the intention to leave the profession, 24% and 27% than the intention to quit the job, 37% and 35%, respectively. Moreover, Ulupinar et al. (2022) conducted a rapid

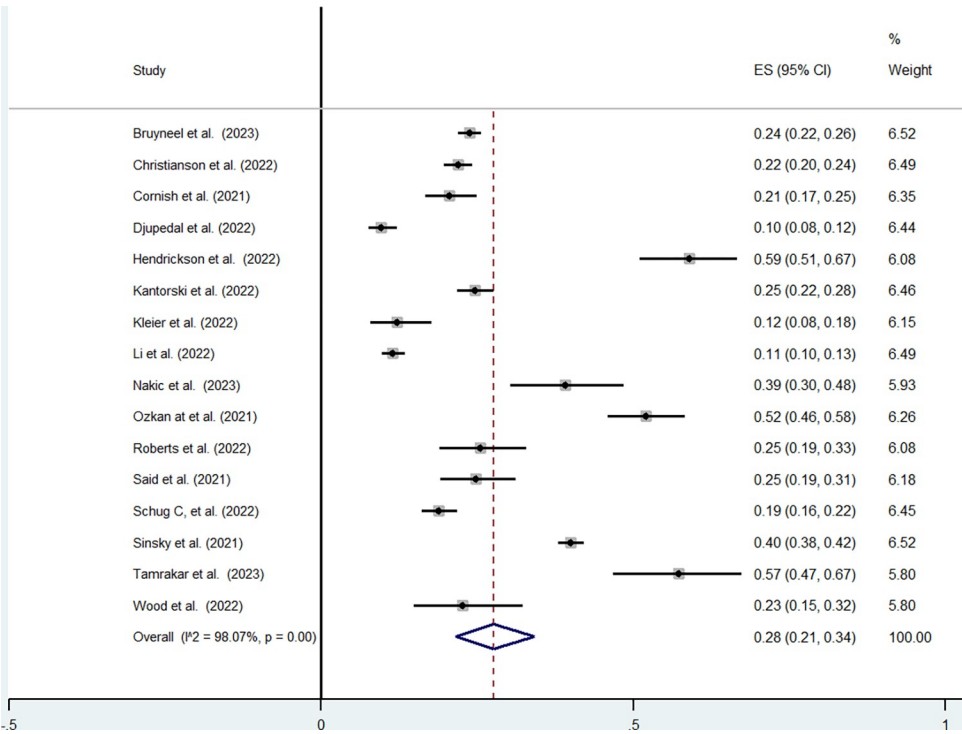

**Fig 4. Nurses' intention to leave the profession.**

meta-analysis when the pandemic was still ongoing. Our meta-analysis started including papers after the COVID-19. This increases the chance of creating a complete overview of prevalence during the pandemic, which may result in different prevalence numbers.

Xu et al. (2023) described a prevalence of intention to leave before the COVID-19 pandemic of 27% for intensive care nurses [131]. These outcomes suggest that the COVID-19 pandemic

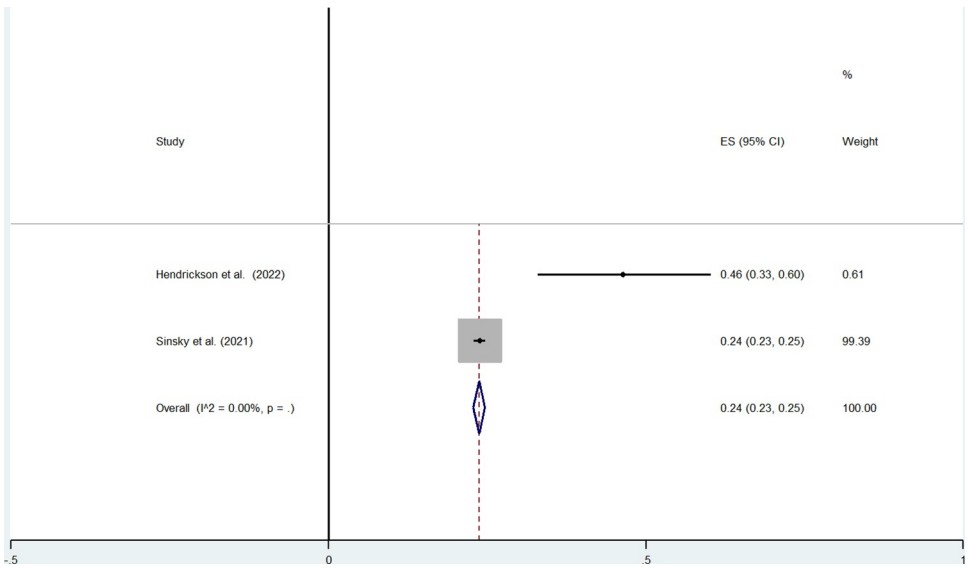

**Fig 5. Physicians' intention to leave the profession.**

increased the intention to quit the job for, by all means, nurses. Similar, comparable literature on physicians in a hospital setting is lacking.

This systematic review revealed burnout symptoms as a primary determinant correlated with the intention to leave during the COVID-19 pandemic. Earlier, conducted systematic reviews showed that epidemics and pandemics have an impact on the mental health of nurses and physicians, resulting in burnout symptoms and PTSD [29, 33]. A systematic review and meta-analysis of qualitative studies by Nasrabadi et al.(2022) showed ICU nurses experienced various psychosocial effects of working during the COVID-19 pandemic, like anxiety, frustration and fear about being infected or infecting one of their loved ones [32]. Additionally, multiple systematic reviews described the high burnout symptoms of nurses and physicians, which may result in the intention to leave [17]. The fear and anxiety going along with the COVID-19 pandemic could affect the stress level of nurses and physicians and can evolve into burnout symptoms. It seems reasonable that determinants like fear, anxiety, burnout symptoms, and PTSD have intercorrelation and can strengthen their impact on the intention to leave [17, 32].

Moreover, this study revealed again the impact of age and experience on the intention to leave. Previous systematic. In contrast, reviews showed that the youngness of healthcare workers and low working experience can increase their intention to leave, whereas elderly with more working experience decrease the intention to leave [17]. Remarkably, the authors found that both young and older healthcare workers are at risk for the intention to leave during the COVID-19 pandemic [78, 80, 81, 83]. Interestingly, being older is a risk factor during a crisis, whereas it is a protective factor under normal circumstances [17]. It is hypothetic that ageing healthcare workers feel exhausted after years of working in healthcare, and the pandemic may create extra stress. This circumstance can result in earlier retirement determined by the COVID-19 pandemic, confirmed by earlier research [132]. Furthermore, Ní Léime et al. (2021) described that older healthcare workers were worried about their health risks due to COVID-19, which made them decide to retire early [132]. It suggests that the power and impact of aging on the intention to leave may differ from normal circumstances and a crisis.

Moreover, the results in this study of the impact of experience on the intention to leave are heterogeneous. Previous literature described that more experience decreased the intention to leave [17]. The result that less experienced nurses and physicians have a higher intention to leave could be explained by feeling overwhelmed by the new crisis caused by the pandemic [87, 133, 134]. More experienced nurses and physicians have higher intentions to leave during COVID-19 than in normal circumstances, which suggests that nurses and physicians are feeling devastated by possibly never having experienced a crisis like this despite their years of experience. This devastating feeling is described by Kovoor et al. (2022) as the association between seniority and expressing more significant depression during the COVID-19 pandemic [134]. Both cases can result in increasing the intention to leave. This may explain the various outcomes of the effect of experience on the intention to leave during COVID-19.

Our paper has particular strengths, including the many countries studied in the meta-analysis and systematic review. This makes the outcomes generalisable at international level. Furthermore, the study was executed according to a rigorous research protocol as documented in PROSPERO; two independent researchers should conduct the data analysis and quality to decrease the risk of bias.

A potential limitation is the missing longitudinal values in meta-analysis pre- and post-COVID-19. The meta-analysis before and after the pandemic would have made it possible to create an exact comparison of prevalence numbers and give a clear view of the impact of COVID-19 itself. Nevertheless, the comparison with different literature makes it possible to evaluate the impact of COVID-19 on retention rates. In addition, the missing meta-analysis of the determinants impacting the intention to leave is a potential limitation. The measurement

outcomes of the included studies were heterogeneous, which made it impossible to perform a meta-analysis on determinants.

Moreover, the difference in measuring the intention to leave could be a limitation of this study. It resulted in a meta-analysis of the intention to quit the job and the intention to leave the profession. Nevertheless, these different results gave an overview of the specific purpose to leave and explained the situation more clearly. Lastly, the included papers in the meta-analysis and systematic review cover more nurses than physicians in the sampling. This may result in less generalizability for physicians.

## Conclusion

The findings of this paper showed the critical need for hospital managers to address the concerning increase in nurses' and physicians' intentions to leave during the COVID-19 pandemic. This intention to leave is affected by a complex conjunction of multiple determinants, including the fear of COVID-19 and the confidence in and availability of PPEs. Moreover, individual factors like age, experience, burnout symptoms, and support are maintained in this review. By implementing effective measures affecting these determinants, hospitals can create a committed and powerful ambience. Resulting in the retention of experienced healthcare personnel, attracting new nurses and physicians, and ultimately increasing the quality of care.

Approaching the lessons learned during the COVID-19 pandemic, it has been highlighted that the well-being of nurses and physicians affects the healthcare system. Addressing the determinants impacting the intention to leave in this review can be the start of a more sustainable healthcare workforce in hospitals in future crises, shouldered by all stakeholders in the healthcare workforce.

## Supporting information

**S1 Table. PRISMA checklist.**
(DOCX)

**S2 Table. Literature search.**
(DOCX)

**S3 Table. Data-extraction and quality assessment summary of included records for systematic review.**
(DOCX)

## Acknowledgments

We want to thank the partners and project staff of the METEOR project for their assistance.

## Author Contributions

**Conceptualization:** Neeltje de Vries, J Peter de Winter.

**Data curation:** Neeltje de Vries, José Bouman.

**Formal analysis:** Neeltje de Vries, Laura Maniscalco.

**Investigation:** Neeltje de Vries, Laura Maniscalco.

**Methodology:** Neeltje de Vries, Laura Maniscalco, J Peter de Winter.

**Project administration:** Neeltje de Vries.

**Resources:** Neeltje de Vries, José Bouman.

**Supervision:** Domenica Matranga, J Peter de Winter.

**Validation:** Neeltje de Vries, Laura Maniscalco, Domenica Matranga, J Peter de Winter.

**Visualization:** Neeltje de Vries.

**Writing – original draft:** Neeltje de Vries, Laura Maniscalco.

**Writing – review & editing:** Neeltje de Vries, Laura Maniscalco, Domenica Matranga, J Peter de Winter.

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
