## [Decision Letter · Decision Letter 0]

28 Dec 2023

PONE-D-23-37725Determinants of intention to leave among nurses and physicians in a hospital setting during the COVID-19 pandemic: A systematic review and meta-analysis.PLOS ONE

Dear Dr. de Winter,

Thank you for submitting your manuscript to PLOS ONE. After careful consideration, we feel that it has merit but does not fully meet PLOS ONE’s publication criteria as it currently stands. Therefore, we invite you to submit a revised version of the manuscript that addresses the points raised during the review process.

We look forward to receiving your revised manuscript.

Kind regards,

Fatma Refaat Ahmed, Ph.D.

Academic Editor

PLOS ONE

2. Please include your tables as part of your main manuscript and remove the individual files. Please note that supplementary tables (should remain/ be uploaded) as separate ""supporting information"" files".

Reviewers' comments:

Reviewer's Responses to Questions

**Comments to the Author**

1. Is the manuscript technically sound, and do the data support the conclusions?

Reviewer #1: Yes

Reviewer #2: Yes

2. Has the statistical analysis been performed appropriately and rigorously? 

Reviewer #1: Yes

Reviewer #2: Yes

3. Have the authors made all data underlying the findings in their manuscript fully available?

Reviewer #1: Yes

Reviewer #2: Yes

4. Is the manuscript presented in an intelligible fashion and written in standard English?

Reviewer #1: Yes

Reviewer #2: Yes

5. Review Comments to the Author

Reviewer #1: Dear authors

Thank you for your efforts to do this research. Please See my comments in below.

General Comment:

The manuscript should be edited by a person who fluent in English language.

Additional Comments:

Abstract:

Conclusions: in the sentence" …including the fear of COVID-19 and the confidence in and availability of PPEs."; Please write the term "PPE" in full. Because it is mentioned for the first time.

The number of keywords is high. I suggest reducing them to a maximum of 5 words.

Introduction section:

I suggest that the introduction be developed using a related literature review. For example, authors can use the following articles:

https://www.frontiersin.org/articles/10.3389/fpubh.2022.1034624/full

https://www.mdpi.com/1660-4601/18/23/12548

Method section:

In my opinion, the authors should consider studies published in Scopus or Web of Science databases for inclusion in this manuscript to ensure full coverage of studies.

Authors stated, "Papers were excluded if full text was not available. Moreover, systematic reviews, study protocols, and dissertations were excluded." I recommend qualitative studies be added to the exclusion criteria section.

I have a question: Is the quantitative section of mixed method studies included in the review? If they were included, please add to the included criteria.

I have a question: Is the quantitative section of mixed method studies included in the review? If they were included, please add to the included criteria because, in the results section, it was written that there were three mixed-method designs.

Results section:

Authors stated: "Three mixed-methods designs are included." but in the table "Data-extraction and quality assessment summary of included records for systematic review." it wasn't seen.

In Fig 1. PRISMA flow diagram: Please use the better term instead of "wrong" in phrases of "wrong population", "wrong study design", "wrong outcomes", and "wrong determinant".

Introduction section:

I suggest that the discussion section be developed using a related literature review. For example, authors can use the following articles:

https://www.frontiersin.org/articles/10.3389/fpubh.2022.1034624/full

https://www.mdpi.com/1660-4601/18/23/12548

Best Regards

Reviewer #2: Thank you for giving me the opportunity to review this systematic review and metanalysis about the intention to leave of healthcare workers working in hospital setting during Covid-19 pandemic.

The systematic review presented in this paper identifies six key themes of determinants impacting nurses' and physicians' intention to leave during the COVID-19 pandemic, shedding light on critical factors affecting retention.

The meta-analysis results reveal a substantial prevalence of intention to leave among healthcare professionals, highlighting the urgent need for hospital administrators to address this concerning trend.

This study underscores the multifaceted nature of determinants influencing healthcare workers' retention, emphasizing the role of fear of COVID-19, personal characteristics, and organizational support in shaping their career intentions.

The findings presented in this paper provide valuable insights for developing targeted strategies to retain nurses and physicians in hospital settings during the ongoing pandemic, with a focus on addressing both systemic and individual factors.

Methodologically the paper respects the PRISMA Checklist.

Please correct minor typo in the abstract: ‘’ Fourty-seven’’ and ‘’Metaanalysis’’

P 20 This increases the chanhe of creating

And please check the references, some of them used incorrect abbreviation of journal title.

5. Boniol M, Kunjumen T, Nair TS, et al. The global health workforce stock and distribution in 2020 and 2030: a threat to equity and ‘universal’ health coverage? BMJ Glob Heal;

[29] Alotibi, AM. Alalwan, FH. Alabdallah A. Factors Associated with Nurses Intention to Leave During COVID-19Pandemic: Literatue Review. Saudi J Nurs Heal Care 2023; 6: 159–167.

[60] Petrișor C, Breazu C, Doroftei M, et al. Association of Moral Distress with Anxiety, Depression, and an Intention to Leave among Nurses Working in Intensive Care Units during the COVID-19 Pandemic. Healthc (Basel, Switzerland); 9. Epub ahead of print October 2021. DOI: 10.3390/healthcare9101377.

[107] Jing C, Feng-Hong Z, Yi-Yan W. An investigation of the incidence of post-traumatic stress disorder, turnover intention and psychological resilience among medical staff in a public hospital in China during the outbreak of the omicron variant in the COVID-19 pandemic in2022. Front psychiatry 2022; 13: 999870.

6. PLOS authors have the option to publish the peer review history of their article (what does this mean?). If published, this will include your full peer review and any attached files.

Reviewer #1: No

Reviewer #2: No

---

## [Author Response · Author response to Decision Letter 0]

7 Feb 2024

Rebuttal Manuscript entitled “Determinants of intention to leave among nurses and physicians in a hospital setting during the COVID-19 pandemic: A systematic review and meta-analysis” 

Haarlem, February 2, 2024

Dear Editor,

We want to express our sincere gratitude for the opportunity to revise our manuscript titled “Determinants of Intention to Leave Among Nurses and Physicians in a Hospital Setting During the COVID-19 Pandemic: A Systematic Review and Meta-Analysis.” We sincerely appreciate the insightful comments and constructive suggestions provided by the reviewers. Their feedback has been invaluable in enhancing the quality and clarity of our manuscript.

In response to the reviewers' comments, we have carefully revised our manuscript, making all the necessary amendments. Changes made to the manuscript are highlighted within the text to facilitate the review process. We believe that these revisions have significantly improved our manuscript, making it more comprehensive and relevant to the field.

We are committed to providing any further information or clarification that may be required. Should you have any additional questions or comments, please do not hesitate to contact us. We eagerly await your feedback regarding our submission.

Thank you once again for your cooperation and guidance throughout this process.

Warm regards,

The Authors

 

Reviewers' comments:

Reviewer's Responses to Questions

Comments to the Author

1. Is the manuscript technically sound, and do the data support the conclusions?

Reviewer #1: Yes

Reviewer #2: Yes

2. Has the statistical analysis been performed appropriately and rigorously? 

Reviewer #1: Yes

Reviewer #2: Yes

3. Have the authors made all data underlying the findings in their manuscript fully available?

Reviewer #1: Yes

Reviewer #2: Yes

4. Is the manuscript presented in an intelligible fashion and written in standard English?

Reviewer #1: Yes

Reviewer #2: Yes

5. Review Comments to the Author

Reviewer #1: Dear authors

Thank you for your efforts to do this research. Please See my comments in below.

General Comment:

1. The manuscript should be edited by a person who fluent in English language.

Before submission, the submission was checked by an English native speaker. 

Additional Comments:

Abstract:

1. Conclusions: in the sentence" …including the fear of COVID-19 and the confidence in and availability of PPEs."; Please write the term "PPE" in full. Because it is mentioned for the first time.

We want to thank the reviewer for the essential suggestions. We write the term "PPE" in full: 

“This intention to leave is affected by a complex conjunction of multiple determinants, including the fear of COVID-19 and the confidence in and availability of personal protective equipment”. 

2. The number of keywords is high. I suggest reducing them to a maximum of 5 words.

The number of keywords was reduced to five words: 

“COVID-19, turnover intention, determinants, healthcare workers, meta-analysis”. 

Introduction section:

3. I suggest that the introduction be developed using a related literature review. For example, authors can use the following articles:

https://www.frontiersin.org/articles/10.3389/fpubh.2022.1034624/full

https://www.mdpi.com/1660-4601/18/23/12548

The paper of Narabadi et al.(2022) was adjusted in the introduction section, previous research section: 

“Similar studies have been published in the past years. Nevertheless, these papers, among other things, focused on nurses only[6, 29-32], measured mental issues[29,33,34] increasing the intention to leave, examined the impact of viral outbreaks instead of COVID-19[35] or lacked a systematic approach[36].”. 

The paper of Maleki et al.(2021) (https://www.mdpi.com/1660-4601/18/23/12548) was added in the introduction section:

“After that, the COVID-19 pandemic put severe pressure on the already vulnerable healthcare workforce. Maleki et al. (2021) described that pandemics, like COVID-19, can put a severe burden on the healthcare system[14]. Nurses and physicians were exposed to acute stress[14–16] and high workloads[16]. Previous research showed that workload, stress and burnout symptoms increase the intention to leave[17]. It is essential to focus on retaining healthcare personnel. In contrast, nurses and physicians leaving their jobs can create severe financial and non-financial burdens to the healthcare system[18], e.g. unsatisfactory quality of care[19], recruiting costs[20–23] and dissatisfaction with patients and personnel[20, 24, 25]. Especially during this pandemic crisis, the burden on nurses and physicians was severe[14]. Future pandemics or epidemics can significantly impact the healthcare system and, thus, the outflow of healthcare workers. Moreover, it is crucial to understand how hospital managers can manage to retain their nurses and physicians during future outbreaks.”

Method section:

1. In my opinion, the authors should consider studies published in Scopus or Web of Science databases for inclusion in this manuscript to ensure complete coverage of studies.

We want to thank the reviewer for the vital suggestion. To ensure a full coverage of studies, the Web of Science databases was added in the search. 

“The authors developed an extensive search string for three scientific databases: PubMed, Embase, CINAHL, and Web of Science.”

This search resulted in extra included papers in our manuscript: 

“The literature search resulted in 2261 papers. Before screening, 674 duplicates were removed. After title and abstract screening, 1045 records were removed due to not fulfilling the inclusion criteria. The full text of 158 records was screened. Finally, 55 papers fulfilled the eligibility criteria and were included in the systematic review and of these, 33 papers could be included in the meta-analysis. The study selection process is shown in the flow chart (Fig 1).”. 

Within the manuscript, the newly included papers are described in Table 1, the text of the systematic review part and S3_Table. 

2. Authors stated, "Papers were excluded if full text was not available. Moreover, systematic reviews, study protocols, and dissertations were excluded." I recommend qualitative studies be added to the exclusion criteria section.

Thanks to the reviewer for this recommendation. The method section was clarified: 

“Papers were excluded if full text was not available. Moreover, qualitative studies, systematic reviews, study protocols, and dissertations were excluded.”. 

3. I have a question: Is the quantitative section of mixed method studies included in the review? If they were included, please add to the included criteria.

To ensure that we included all possible input from research, we included the quantitative section of mixed method studies. This is clarified in the methodology section:

“Quantitative research and mixed methods were included in the manuscript. Only the quantitative part of the mixed-method studies will be included.”. 

4. I have a question: Is the quantitative section of mixed method studies included in the review? If they were included, please add to the included criteria because, in the results section, it was written that there were three mixed-method designs.

See the comment above. The method section was adjusted to clarify the following:

“Quantitative research and mixed methods were included in the manuscript. Only the quantitative part of the mixed-method studies will be included.”.

Results section: 

5. The authors stated: "Three mixed-methods designs are included." but in the table "Data-extraction and quality assessment summary of included records for systematic review." it wasn't seen.

Thank you for the suggestion. However, three mixed-method design studies are included in the meta-analysis part. They are thus shown in Table 1 “Table 1. Data-extraction table and quality assessment summary of included records in meta-analysis”. The three included papers are Crowe et al.(2022), Wood et al.(2021) and Wood et al. (2022).

6. In Fig 1. PRISMA flow diagram: Please use the better term instead of "wrong" in phrases of "wrong population", "wrong study design", "wrong outcomes", and "wrong determinant".

Thank you for the suggestion. We modify the term “wrong” with “different”. 

Discussion section:

7. I suggest that the discussion section be developed using a related literature review. For example, authors can use the following articles:

https://www.frontiersin.org/articles/10.3389/fpubh.2022.1034624/full

https://www.mdpi.com/1660-4601/18/23/12548

Whereas the paper of Maleki et al.(2021) did better fit the introduction section instead of the paragraph in our discussion section, we chose to include the paper in the introduction (See comment three above). 

However, we amplified the discussion section with the exciting paper of Nasrabadi et al.(2022): 

“This systematic review revealed burnout symptoms as a primary determinant correlated with the intention to leave during the COVID-19 pandemic. Earlier conducted systematic reviews showed that epidemics and pandemics have an impact on the mental health of nurses and physicians, resulting in burnout symptoms and PTSD[29, 33]. A systematic review and meta-analysis of qualitative studies by Nasrabadi et al.(2022) showed ICU nurses experienced various psychosocial effects on working during the COVID-19 pandemic, like anxiety, frustration and fear about being infected or infecting one of their loved ones[32]. Additionally, multiple systematic reviews described the high burnout symptoms of nurses and physicians, which may result in the intention to leave[17]. The fear and anxiety going along with the COVID-19 pandemic could affect the stress level of nurses and physicians and can evolve into burnout symptoms. It seems reasonable that determinants like fear, anxiety, burnout symptoms, and PTSD have intercorrelation and can strengthen their impact on the intention to leave[17, 32].”. 

Reviewer 2:

Thank you for giving me the opportunity to review this systematic review and meta-analysis about the intention to leave healthcare workers working in hospital settings during the COVID-19 pandemic.

The systematic review presented in this paper identifies six key themes of determinants impacting nurses' and physicians' intention to leave during the COVID-19 pandemic, shedding light on critical factors affecting retention.

The meta-analysis results reveal a substantial prevalence of intention to leave among healthcare professionals, highlighting the urgent need for hospital administrators to address this concerning trend.

This study underscores the multifaceted nature of determinants influencing healthcare workers' retention, emphasising the role of fear of COVID-19, personal characteristics, and organisational support in shaping their career intentions.

The findings presented in this paper provide valuable insights for developing targeted strategies to retain nurses and physicians in hospital settings during the ongoing pandemic, with a focus on addressing both systemic and individual factors.

Methodologically, the paper respects the PRISMA Checklist.

Dear reviewer, We would like to thank the reviewer for the beautiful words spent on our paper. We are delighted that you enjoyed our work.

1. Please correct minor typos in the abstract: ‘’ Fourty-seven’’ and ‘’Metaanalysis.’’

P 20 This increases the chance of creating

The first typo in the abstract was removed because the number of included papers in the systematic review was adjusted. This resulted in the following sentence: 

“Fifty-five papers were included in the systematic review, whereas thirty-three papers fulfilled the eligibility criteria for the meta-analysis.”.

We have adjusted the typo in the text: 

“Secondly, a meta-analysis on the prevalence of intention to leave for nurses and physicians during the COVID-19 pandemic”. 

“This increases the chance of creating a complete overview of prevalence during the pandemic, which may result in different prevalence numbers.”. 

2. please check the references; some of them used incorrect abbreviations of journal titles.

5. Boniol M, Kunjumen T, Nair TS, et al. The global health workforce stock and distribution in 2020 and 2030: a threat to equity and ‘universal’ health coverage? BMJ Glob Heal;

[29] Alotibi, AM. Alalwan, FH. Abdallah A. Factors Associated with Nurse's Intention to Leave During COVID-19 Pandemic: Literature Review. Saudi J Nurs Heal Care 2023; 6: 159–167.

[60] Petrișor C, Breazu C, Doroftei M, et al. Association of Moral Distress with Anxiety, Depression, and an Intention to Leave among Nurses Working in Intensive Care Units during the COVID-19 Pandemic. Healthc (Basel, Switzerland); 9. Epub ahead of print October 2021. DOI: 10.3390/healthcare9101377.

[107] Jing C, Feng-Hong Z, Yi-Yan W. An investigation of the incidence of post-traumatic stress disorder, turnover intention and psychological resilience among medical staff in a public hospital in China during the outbreak of the omicron variant in the COVID-19 pandemic in 2022. Front Psychiatry 2022; 13: 999870.

Thank you for this observation. We double-checked and modified the abbreviations in the references. 

“Boniol M, Kunjumen T, Nair TS, et al. The global health workforce stock and distribution in 2020 and 2030: a threat to equity and ‘universal’ health coverage? BMJ Global Health; 7. Epub ahead of print June 2022. DOI: 10.1136/bmjgh-2022-009316.”

“Alotibi, AM. Alalwan, FH. Alabdallah A. Factors Associated with Nurses Intention to Leave During COVID-19Pandemic: Literatue Review. Saudi J Nurs Health Care 2023; 6: 159–167.”

“Petrișor C, Breazu C, Doroftei M, et al. Association of Moral Distress with Anxiety, Depression, and an Intention to Leave among Nurses Working in Intensive Care Units during the COVID-19 Pandemic. Healthcare (Basel, Switzerland); 9. Epub ahead of print October 2021. DOI: 10.3390/healthcare9101377.”

“Jing C, Feng-Hong Z, Yi-Yan W. An investigation of the incidence of post-traumatic stress disorder, turnover intention and psychological resilience among medical staff in a public hospital in China during the outbreak of the omicron variant in the COVID-19 pandemic in 2022. Frontiers in Psychiatry 2022; 13: 999870.”.

---

## [Decision Letter · Decision Letter 1]

27 Feb 2024

Determinants of intention to leave among nurses and physicians in a hospital setting during the COVID-19 pandemic: A systematic review and meta-analysis.

PONE-D-23-37725R1

Dear Dr. de Winter,

We’re pleased to inform you that your manuscript has been judged scientifically suitable for publication and will be formally accepted for publication once it meets all outstanding technical requirements.

Kind regards,

Fatma Refaat Ahmed, Ph.D.

Academic Editor

PLOS ONE

Reviewers' comments:

Reviewer's Responses to Questions

**Comments to the Author**

1. If the authors have adequately addressed your comments raised in a previous round of review and you feel that this manuscript is now acceptable for publication, you may indicate that here to bypass the “Comments to the Author” section, enter your conflict of interest statement in the “Confidential to Editor” section, and submit your "Accept" recommendation.

Reviewer #1: All comments have been addressed

Reviewer #2: All comments have been addressed

2. Is the manuscript technically sound, and do the data support the conclusions?

Reviewer #1: Yes

Reviewer #2: Yes

3. Has the statistical analysis been performed appropriately and rigorously? 

Reviewer #1: Yes

Reviewer #2: Yes

4. Have the authors made all data underlying the findings in their manuscript fully available?

Reviewer #1: Yes

Reviewer #2: Yes

5. Is the manuscript presented in an intelligible fashion and written in standard English?

Reviewer #1: Yes

Reviewer #2: Yes

6. Review Comments to the Author

Reviewer #1: Dear authors

Thank you for addressing my comments. I have two comments as follows:

The clarity of the figures should be improved.

In Table 1, the third column (country) should be merged with the first column (first author and year), as well as column 4 (sample) with the fifth column (sample size).

Best regards

Reviewer #2: The authors have address all my comments and corrected the references and some minor spelling mistake, in my opinion the paper is ready to be publish.

7. PLOS authors have the option to publish the peer review history of their article (what does this mean?). If published, this will include your full peer review and any attached files.

Reviewer #1: No

Reviewer #2: No

---

## [Editor Report · Acceptance letter]

5 Mar 2024

PONE-D-23-37725R1 

PLOS ONE

Dear Dr. de Winter, 

I'm pleased to inform you that your manuscript has been deemed suitable for publication in PLOS ONE. Congratulations! Your manuscript is now being handed over to our production team.

Kind regards, 

on behalf of

Dr. Fatma Refaat Ahmed 

Academic Editor

PLOS ONE